# Interval price prediction of livestock product based on fuzzy mathematics and improved LSTM

**Weimin Ma[1], Lingling Peng[1,2]\*, Hu Chen[1], Haisheng Yan[2]**

**1** School of Economics and Management, Tongji University, Shanghai, China, **2** Chongqing University of Arts and Sciences, Chongqing, China

\* llpeng@tongji.edu.cn

## Abstract

Livestock product prices serve as a barometer and bellwether for the agricultural market. However, traditional point prediction techniques focus mainly on tracking or fitting, resulting in limited information and challenges in evaluating the uncertainty of future prices. A comprehensive livestock price prediction model with joint point and interval prediction capabilities is proposed, with fuzzy mathematics and long short-term memory. Three main steps are taken: (1) data composition and reconstruction, to extract a set of relatively stationary subsequence components by complementary ensemble empirical mode decomposition (CEEMD) from original signal, and divide these components into three groups according to fuzzy entropy (FE) value. (2) characteristics categorization, determining the lower bound, mean, and upper bound of the rebuilt data via fuzzy information granulation (FIG) to better characterize the price fluctuation range. (3) price prediction, including point and interval predictions with attention mechanism long short-term memory (AM-LSTM). An empirical study was conducted on the weekly price data of pork, beef, and mutton in China from 2009 to 2023, incorporating discussions on different embedding dimensions, prediction step, fuzzy granulation window sizes, decomposition techniques, and prediction algorithms. The results indicate that the proposed interval prediction model can not only achieve high accuracy in point prediction, but also better capture price change intervals.

## 1. Introduction

Livestock products, as a vital component of agricultural products, are important for improving human physical fitness, local population diet habits, agriculture's internal structure, and the nation's economy [1–3]. China is currently experiencing a surge in the development of modern husbandry, enthusiastically establishing a long-term mechanism for the stable and orderly development of pig products, and actively promoting large-scale standardized and mechanized breeding. In 2023, China produced a total of 96.41 million tons of pigs, cows, sheep, and poultry, an increase of 4.5% compared to the previous year. In recent years, under the combined impact of the continuous upgrading of residents' consumption patterns and unstable supply of livestock products, the market prices of major livestock products such as pork, beef, and mutton have fluctuated frequently, which has had adversely affected the formation of consistent

**Data availability statement:** All data files are available from the CEIC database (https://doi.org/10.3886/E211561V1)

**Funding:** The author(s) received no specific funding for this work.

expectations for production and consumption. Therefore, as smart agriculture rapidly evolves, the investigation on the volatility and forecasting of livestock product price through technologies such as the Internet of Things, big data, and artificial intelligence has become an essential topic for scholars both domestically and internationally [4].

The fluctuation of livestock product prices exhibits high nonlinearity and instability, being intricately influenced by a multitude of complex factors such as market supply and demand, production costs, and policy regulations. This results in a high degree of complexity and uncertainty in the related data, posing a significant challenge for accurate prediction [5]. Statistical and artificial intelligence techniques are the two main approaches used to anticipate the products price [6–8]. auto regressive moving average (ARMA) Traditional statistical methods have difficulty with the highly nonlinear and unstable nature of animal commodity price data series. Data-driven models based on machine learning (ML) have emerged for agricultural product prices prediction thanks to their excellent mapping and self-learning capabilities. Many research has applied traditional Machine Learning (ML) models, including autoregressive models, back propagation neural network (BPNN), extreme learning machine (ELM) and random forest, in the field of product price prediction, encompassing livestock products, major crops, and other commodities [9–12]. Jin et al. [13] developed a Gaussian process regression (GPR) model to forecast wholesale prices of yellow corn. Zhou et al. [14] employed an optimized relevance vector machine (RVM) to accurately predict the prices of three precious metals: silver, palladium, and platinum. However, these models often encounter issues such as large prediction errors and insufficient adaptability when dealing with complex dynamic changes and large-scale datasets. Given that product prices evolve dynamically over time, there is a complex interplay between current prices and historical time series data, which profoundly reveals the causality and correlation between time series. To effectively capture these complex relationships, deep learning (DL) technologies have emerged, such as convolutional neural networks (CNN) and long short-term memory (LSTM) models, which can significantly improve the temporal dependence of time series data and capture long-term dependencies within it. Numerous studies have demonstrated the outstanding performance of the DL mode in livestock product price prediction. Nayak et al. [15] had shown that ML algorithms outperform conventional statistical methods in predicting the prices for essential crops like tomatoes, onions, and potatoes in major India markets. Gu et al. [16] proposed an innovative dual input attention LSTM model, which demonstrated high efficiency in predicting agricultural product prices. A Hidden Markov (HM) had been combined with six baseline DL models, Recurrent Neural Networks (RNN), CNN, LSTM, Gated Recurrent Units (GRU), Bidirectional LSTM and Bidirectional GRU to predict the nonlinear and nonstationary price data of agricultural commodities in [17]. Harshith et al. [18] demonstrated that time series analysis models such as Deep Neural Networks (DNN), RNN, LSTM, and GRU, are highly effective in forecasting the daily cumin prices of Unjha market, Gujarat, India. Furthermore, these deep learning techniques are also widely applied in the prediction of intrusion detection systems. These deep learning algorithms, especially LSTM [5], demonstrate robust capabilities in time series prediction, capable of finely interpreting temporal information and utilizing it to identify deep causal relationships hidden within data sequences. When dealing with datasets that are highly dependent on time, this not only enhances sensitivity to time series features, but also significantly improves the model's predictive power and accuracy by maintaining the temporal integrity of the data and capturing subtle temporal variations. However, LSTM also exhibits strong dependencies on feature selection and the configuration of model hyperparameters. Many meta-heuristic algorithms, such as particle swarm optimization (PSO) [19], crow search algorithm (COA) [5], and differential evolution (DE) [20], are often used to optimize LSTM for hyperparameter tuning, weight initialization, and other tasks. Yet, when

processing large-scale datasets, these algorithms face challenges such as high computational resource consumption, complex parameter adjustment, and susceptibility to local optimization. The introduction of attention mechanisms (AM) in LSTM models [21], however, can dynamically focus on key information within the sequence, enhancing the model's attention to important features and improving the accuracy and flexibility of sequence modeling and prediction. This innovation has garnered widespread attention in the field of optimization. In order to increase the flexibility and precision of time series prediction models in a diverse data source environment, data decomposition technology has emerged as a crucial tactic. This approach reduces the difficulty of prediction tasks by using preprocessing of complicated and fluctuating time series to uncover simpler and more readable signal patterns within the data [22,23]. Data decomposition techniques mainly include wavelet packet transform [24], empirical mode decomposition (EMD) [25–27], variational mode decomposition (VMD) [28], and other variants. Fu et al. [29] provided a robust prediction approach for pig prices based on ensemble empirical mode decomposition (EEMD) and LSTM. Li et al. [30] adopted a combination of the VMD and GRU neural network to predict the prices of beef, mutton, and pork while taking into account the effects of heterogeneous non-time series on livestock products price, such as the variety, growth cycle, longitude and latitude. Sun et al. [31] introduced a comprehensive prediction model that combined VMD, EEMD, and LSTM to forecast the average wholesale weekly price for pork, Chinese chives, shiitake mushrooms, and cauliflower. Out of all the data decomposition techniques, complementary ensemble empirical mode decomposition (CEEMD) introduces positive and negative paired noise, effectively reduces mode aliasing and endpoint effects, and exhibits excellent adaptability, hence greatly improving the stability and accuracy of decomposition. Predicting precipitation, wind speed, wave height, and other variables is another common application for CEEMD.

However, we are aware that training time increases along with feature space dimension as machine learning progresses, which eventually results in the "curse of dimensions" issue. Consequently, various methods for data dimensionality reduction have emerged. The fuzzy entropy (FE), as a fundamental principle in fuzzy information theory, allows for the quantification of complexity and uncertainty with fuzzy systems, which is an improvement over sample entropy (SE) and approximate entropy (AE) [32–34]. Evaluating the complexity of the information derived from data decomposition in all frequency bands—from low frequency to high frequency—is crucial to lowering the number of sub-prediction models and improving overall prediction performance. In order to reconstruct the new components, neighboring components with comparable entropy values are stacked after fuzzy entropy values are computed on several frequency scales. In light of the aforementioned circumstance, the combination of deep learning models including LSTM and attention with data combination techniques is a cutting-edge method for time series prediction.

The majority of the previously examined literature concentrates on point predictions, or the single value of future prices for animal products, which is insufficient to fully capture the volatility and unpredictability of future price swings, and may be more susceptible to external shocks such as anomalies and atypical events. Interval prediction is a great way to deal with these problems, in addition to single-value point prediction, it provides more detailed information about the data development process, including average trend, maximum and lowest values. This implies that interval prediction reduces noise interference, yields more solid and dependable prediction findings, and offers a more detailed understanding of price fluctuations and market dynamics. Interval prediction is categorized into two primary categories: indirect and direct methods. The direct interval prediction approach adopts statistical theories like delta [35], Bayesian [36], mean square error estimation (MVE) [37], bootstrap [38], and construct interval prediction models involving

multivariate grey models [39], interval fuzzy rule model [40] or vector error correction model [41] to estimate the upper and lower bounds of interval data as a whole. Alternatively, it makes these assumptions based on extensive calculations. For the indirect interval prediction method, the upper and lower bounds of interval data are considered as two independent point value sequences for prediction, and the prediction results of the two sequences are then utilized to determine the prediction interval. Zhang et al. [42] suggested a highly competitive model for predicting pork price intervals by coupling vector error correction model (VECM) with an artificial intelligence model (Coin AIs) that considered the cointegration connections of lower and upper bound. Wang et al. [43] introduced the multi-objective grey wolf optimizer (MOGWO) optimized by BPNN to establish upper and lower bounds based on point prediction results, achieving exceptional coverage and accuracy in interval prediction. To establish fuzzy inference systems based on short-term prediction models, Li et al. [44] recommended a novel association rule for information granules that may extract and derive the correlation between two trend feature sets relating to the past and future. The fuzzy information granules mentioned above, as an application extension of fuzzy mathematics, enables direct partition granular division on large-scale time series, without requiring a prior determination of priority distribution. Specifically, the fuzzy information granulation (FIG) technique decomposes the original time series into individual time windows, with each window being represented by a fuzzy set that holistically reflects the range, mean, and trend characteristics, which eases the comprehension and interpretation of data changes within each time window. Finally, time series analysis shifts from a numerical platform to a newly constructed granularity platform, which not only enables the discovery of comprehensive and interpretable abstract knowledge but also effectively reduces the dimensionality of complex problems, while minimizing computational overhead as much as possible. Fuzzy reasoning in conjunction with neural networks, support vector machines, and other artificial intelligence algorithms have become a critical research field in recent years. Xia et al. [45] developed a hybrid framework by combining fuzzy information granulation algorithm, improved variational mode decomposition technique, and high-order fuzzy cognitive map, to realize interval prediction of photovoltaic power generation. Zhu et al. [46] employed used multi linear trend FIG and LSTM frameworks to predict the trend changes, fluctuation magnitudes, and trend persistence of long-term time series. Liu et al. [47] developed a new interval long-term prediction model developed by integrating trend-based information granules (TIG), fuzzy time series, and ensemble learning. In conclusion, interval prediction based on FIG can offer a solution for anticipating long-term future data and communicating the uncertainty of data in the future, as opposed to traditional numerical time prediction.

To thoroughly assess the point and interval prediction results, this study conducts empirical research based on the prices of popular livestock and poultry products in China, introduces a novel heterogeneous interval price prediction model, and compares it with both filtered and unfiltered models with a variety of evaluation indicators. It fully utilizes the strengths of CEEMD, FE, FIG and attention mechanism LSTM, while also considering the crucial importance of embedding dimension, prediction step, fuzzy granulation window size, decomposition method, and prediction algorithm in defining product price, to demonstrate the scientific approach and forecast accuracy of the model. The main contributions of this novel method can be illustrated as follows: (1) FIG is introduced for the first time to quantify the uncertainty in livestock price prediction, combines LSTM with attention mechanism to estimate the lower and upper bound of future livestock price, allowing us to take full of machine learning's ability to improve interval forecasting performance, which is critical for managing livestock prices. (2) The proposed prediction

framework can provide accurate point prediction and reliable interval prediction, and compared with other machine learning methods, it performs exceptionally well in a variety of prediction scenarios by fully utilizing advanced data processing techniques and algorithm optimization, which includes, but is not limited to, higher accuracy, stronger generalization ability, and quick adaptation to new data. (3) Two different metrics are implemented to evaluate the entire interval-valued forecasting performance: one is a standard measure for forecast performance (i.e., conventional mean absolute percentage error, MAPE, and root mean square error, RMSE); and the other is a unique measure for interval-valued data (i.e., forecasting interval coverage percentage, FICP, and forecasting interval average width, FIAW).

The rest of the study is as follows. Section 2 contains the theoretical methods. Section 3 introduces the framework of the price forecasting model. Section 4 displays the empirical analysis of the price prediction of pork, beef and mutton in China. Section 5 contains the main conclusions and the prospects.

## 2. Methods

### 2.1. CEEMD

EMD technique improves the accuracy of predicting economic events like price swings of livestock products and data interpretability by breaking down nonlinear and non-stationary time series into several Intrinsic Mode Functions (IMFs) and exposing the data's intrinsic periodic properties [48]. Yeh et al. proposed the CCEMD, which introduced complementary noise that is entirely negatively correlated, independently and identically distributed, to overcome the potential mode mixing problem during EMD process [49]. The decomposition process of CEEMD can be expressed as:

$$x(t) = \sum_{i=1}^{n} C_i(t) + r_n(t) \tag{1}$$

Where $x(t)$ represents the original signal, $C_i(t)$ is the i-th IMF component, and $r_n(t)$ stands for the residual component. In the CEEMD, the original signal undergoes a preprocessing step where pairs of positive and negative Gaussian white noise, denoted by $u^+$ and $u^-$, respectively, are added to generate positive and negative sequences.

$$\begin{aligned} x_i^+(t) &= x(t) = u^+ \\ x_i^-(t) &= x(t) = u^- \end{aligned} \tag{2}$$

Subsequently, EMD decomposition is applied to each of these noise-added sequences, corresponding IMF components and residual components are obtained. The final result of the CEEMD decomposition is as follows:

$$\begin{aligned} IMF_j(t) &= \frac{1}{2} \sum_{i=1}^{n} IMF_{ij}(t) \\ r(t) &= \frac{1}{2} \sum_{i=1}^{n} r_i(t) \end{aligned} \tag{3}$$

By adopting this method, CEEMD successfully minimizes noise interference during decomposition and enhances the physical significance and interpretability of the IMFs. This study sets the chosen iteration number M to 200 and the white noise amplitude k to 0.2 based on the features of the research data.

## 2.2. FE

FE is an interdisciplinary concept of information theory and fuzzy logic, which are used to quantify the irregularity of nonlinear time series in complex system science and chaos theory. Compared with AE and SE, FE excels in monotonicity and consistency, while describing various signals and embodying robustness against noise. The smaller the entropy value, the simpler the sequence; The larger the entropy value, the more complex the sequence. The fuzzy entropy solution is as follows:

$$FE = \lim_{N \to \infty} (\ln \Phi^m(n,r) - \ln \Phi^{m+1}(n,r)) \tag{4}$$

Where, $\Phi^m$(n,r) calculate from similarity, $m$ is the embedding dimension, $n$ is the gradient for exponential boundary, and $r$ is the similarity tolerance. $m = 2$, $n = 2$ is frequently employed, usually $r = 0.1\sigma_{SD} \sim 0.25\sigma_{SD}$, $\sigma_{SD}$ is the standard deviation of the original sequence. The specific process is detailed in the literature [33].

## 2.3. FIG

L. Zadeh introduced fuzzy set theory in the 1960s and expanded upon in 1979 by proposing the idea of information granules. W. Pedrycz et al. [50] further analyzed that time series can be viewed as a series of fuzzy information particles, each of which is a fuzzy set on the real number line. With the help of this technique, we may examine time series at the granular level to reveal deep and easily understandable patterns.

The FIG construction process essentially comprises two fundamental components: window division and information fuzzification. Window division is the process of partitioning the entire data $X = \{x_1, x_2, …, x_n\}$ into several subsequences $X = \{w_1, w_2, …, w_n\}$. It is advantageous to choose a temporal window width that is reasonable for preserving data information. The process of creating fuzzy particles $P_i$, $P_i = A(w_i)$ which effectively represent the window data for each time window, is known as information fuzzification. Determining function, A, also referred to as the membership function of fuzzy notion, G is the process of fuzzification. Fuzzy particles are frequently shaped as triangular, trapezoidal, Gaussian, parabolic, etc. Obtaining the maximum and minimum values of window data is crucial since the interval prediction in this article mostly depends on the product price fluctuation range. As a result, the triangle model particle is utilized here, and its membership function can be expressed as:

$$A(x,a,m,b) = \begin{cases} 0, & x < a \\ \frac{x-a}{m-a}, & a \le x \le m \\ \frac{b-x}{b-m}, & m \le x \le b \\ 0, & x > b \end{cases} \tag{5}$$

In the formula, the parameters $a$, $m$ and $b$ stand for the minimum (LOW), average (R), and maximum (UP) of the initial data change, respectively. $x$ is a variable within the domain.

## 2.4. AM-LSTM

LSTM effectively solves the problems of gradient vanishing and exploding in traditional recurrent neural networks through its unique gating structure (forget gate, input gate, and output gate), as shown in Fig 1. However, it appears that typical LSTM networks are insufficient to handle the problems of complicated contextual information and long-term dependent relationships. This study proposes an LSTM that combines attention mechanism, called

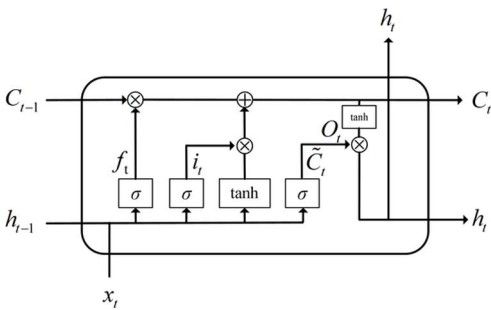

**Fig 1. The internal structure of LSTM unit.**

AM-LSTM. Prediction accuracy is increased by the model's adaptive weighting of the neural network's hidden layer components, which lessens the impact of unimportant parameters and emphasizes the importance of important ones.

$$
\begin{aligned}
f_t &= \sigma(W_f \cdot [h_{t-1}, x_t] + b_f) \\
i_t &= \sigma(W_i \cdot [h_{t-1}, x_t] + b_i) \\
\tilde{C} &= \beta(W_C \cdot [h_{t-1}, x_t] + b_C) \\
C_t &= f_t * C_{t-1} + i_t * \tilde{C}_t \\
Q_t &= \sigma(W_0 \cdot [h_{t-1}, x_t] + b_0) \\
h_t &= O_t * \tanh(C_t)
\end{aligned}
\tag{6}
$$

Where, $x_t$ and $h_t$ represent the input and output information of the hidden layer. $C$ is the memory unit, $\sim C$ is a candidate unit that ultimately determines which information will be added to $C$ through the control of input gates and forget gates. $W$ is the weight parameter matrix, and $b$ is the bias. $\sigma(\cdot)$ and $\beta(\cdot)$ are the sigmoid and tanh activation functions. Attention mechanism can be introduced to improve the LSTM structure and enhance feature extraction efficiency.

Fig 2 shows the attention mechanism process. The fundamental function of an attention mechanism is to create a weight coefficient for each target and multiply it with the input to identify which features in the input are significant to the target and which are not. Firstly, we take into account the raw input data as *<Key, Value>* pairs, and calculate the similarity coefficient between *Key* and *Query*. Based on the *Query* in the given task of the target, we can get the weight coefficient corresponding to *Value*, and then multiply weight coefficient with *Value* to get the output. We use *Q, K*, and *V* to represent *Query, Key*, and *Value*; the formula for calculating the weight coefficient *W* is shown in Equation (7)

$$
W = soft\max(QK^T)
\tag{7}
$$

Equation (8) provides the formula for determining the attention weight coefficient *W* by *Value* to generate the output *a* containing the attention.

$$
a = attention(Q, K, V) = W \odot V = soft\max(QK^T) \odot V
\tag{8}
$$

# 3. Proposed prediction model

The hybrid model described in this work not only predicts the future average product value as a point, but it also predicts the future price fluctuation range as an interval. In light of the benefits of empirical modal decomposition in series smoothing, the advantages of fuzzy information granulation in the granulation interval, and the superior performance of long and short-term memory neural networks in time series data prediction, this paper selects the CEEMD decomposition model, FIG granulation model and AM-LSTM network prediction model, and also adopts the FE algorithm to recombination the components after CEEMD decomposition. Finally, we propose a combined CEEMD-FE-FIG-AM-LSTM prediction model. Fig 3 illustrates the specific steps in this procedure.

(1) Data composition. The initial price sequence $X = \{x_1, x_2, …, x_n\}$, with a length of $N$, can be efficiently decomposed into serval IMF components and one residual component by CEEMD. These components capture the dynamic characteristics of the price series at different scales.

(2) Group allocation. The average fuzzy entropy of each IMF component is computed, serving as a criterion to assess the temporal complexity of the series. Based on this criterion,

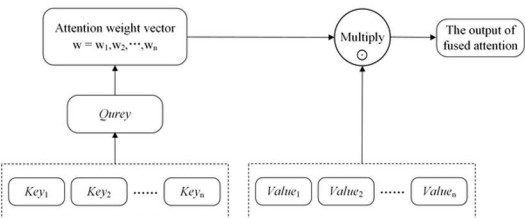

**Fig 2. Diagram of attention structure.**

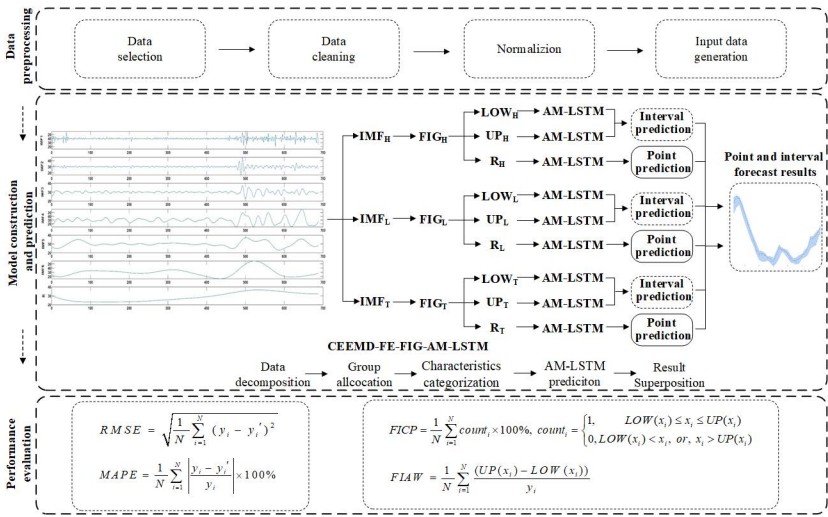

**Fig 3. Structure frame of livestock product price prediction.**

the IMF components are categorized into three groups: the first group includes components with fuzzy entropy higher than a specified threshold, the second group consists of components with fuzzy entropy lower than the threshold, and the third group encompasses the remaining components. By reducing the dimensionality of IMF components, we aim to simplify the construction complexity of the predictive model.

(3) Characteristics categorization. The FIG method is utilized to divide the three reconstructed modal component groups into non-overlapping, fixed-width subsequences. Each subsequence represents a time window, denoted as $w_1, w_2, …, w_t$. Within each time window, effective information is extracted using a triangular membership function and represented by fuzzy particles $G_1, G_2, …, G_t$. These fuzzy particles form three sets of sequences: $\{a_1, a_2, …, a_t\}$, $\{m_1, m_2, …, m_t\}$ and $\{b_1, b_2, …, b_t\}$, representing low (LOW), range (R), and upper bounds (UP) respectively. These sequences reflect the average and dispersion range of price changes, where LOW and UP serve as the lower and upper bounds for interval prediction, and R represents the overall trend of price changes. To explore the influence of time intervals on the prediction system, various time windows will be selected for granulation in subsequent experiments.

(4) AM-LSTM prediction. AM-LSTM model is employed to train and predict for the three modal component groups separately. By incorporating the attention mechanism, the AM-LSTM model can better capture long-term dependencies in the time series data, thereby enhancing prediction accuracy.

(5) Forecast results superposition. For the prediction results of the three modal component groups, the final interval prediction result is obtained by linearly aggregating the LOW and UP sequences of each group, while the aggregation result of the R sequences serves as the final point prediction result. This aggregation strategy not only provides interval predictions for price changes but also offers more precise market trend analysis through point prediction results.

In this paper, the MAPE and RMSE are developed as point forecasting measurement standards to more precisely assess the prediction results of the model through observation and data analysis. But the evaluation criteria MAPE and RMSE, which are typically used for single-value time series, have limitations in evaluating the full interval. Consequently, the FICP and FIAW are added to comprehensively evaluate the upper bound (UP series), average value (R series), and lower bound (LOW series) independently, we should require that: (i) the prediction interval cover the actual observation results as much as possible; and (ii) a narrower prediction interval is more effective than a wider one.

## 4. Experimental results

The experimental environment is Intel(R) Core (TM) i5-8250U, with a 1.80GHz processor. The algorithm model uses MATLAB R2022b as programming language.

### 4.1. Empirical design

**4.1.1. Data.** Based on weekly retail price data of agricultural products monitored from 500 markets across China by the Ministry of Agriculture and Rural Affairs, as sourced from CEIC Database (https://www.ceicdata.com.cn/) and illustrated in Fig 4, the prices of major livestock products in China, encompassing pork, beef, and mutton, have exhibited a notable upward trajectory since the year 2009. Pork prices fluctuate the most among them, frequently exhibiting a high-low volatility phenomenon, has gone through five full pig cycles: April 2003

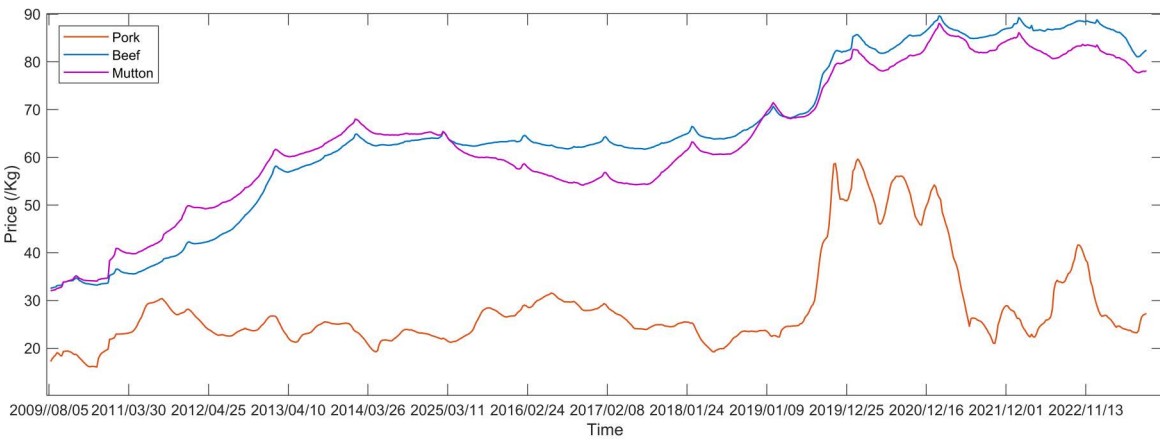

**Fig 4. The weekly price trend of major livestock products since August 2009.**

**Table 1. Basic statistics of weekly price.**

| Dataset | Sample Number | Min (Yuan/kg) | Mean (Yuan/kg) | Max (Yuan/kg) | SD |
|---|---|---|---|---|---|
| Pork | 688 | 16.1 | 28.4 | 59.6 | 9.5 |
| Beef | 688 | 32.5 | 65.2 | 89.6 | 16.4 |
| Mutton | 688 | 32.1 | 63.9 | 88.1 | 14.3 |

to May 2006, June 2006 to May 2010, June 2010 to May 2014, June 2014 to February 2019, and March 2019 to March 2022, and currently, it is remaining in the sixth cycle. The prices of beef and mutton continue to rise but remain relatively stable. Three datasets, each contains 688 weekly data points (from August 5, 2009 to August 30, 2023) related to pork, beef, and mutton in the national market, are chosen as empirical objects for multi-step prediction modeling research in this study, the basic statistics is presented in Table 1. To train and forecast the matching point prediction results and interval prediction range, we use the first 80% of the dataset as a training sample and the remaining 20% as a Testing set.

**4.1.2. Parameter specification.** Three popular forecasting techniques, i.e., BPNN, ELM and traditional LSTM are introduced as significant learning paradigms. To validate the effectiveness of our proposed CEEMD-FE-FIG-AM-LSTM model, two types of models are designed for comparison: unfiltered and filtered models. Specifically, the model that makes predictions directly without any preprocessing is referred to as the unfiltered model, while the model that undergoes CEEMD, FE, or FIG processing before prediction is collectively termed the filtered model. To ensure consistency and comparability in experiments, identical specific techniques and related parameter settings are adopted with different learning paradigms. In BPNN, ReLU is chosen as the activation function, with two hidden layers configured. The first hidden layer consists of 100 neurons, and the second hidden layer contains 50 neurons. The number of iterations is set to 100, and the ADAM algorithm is utilized for optimization. In ELM, the number of hidden nodes is set to 64, and the 'sigmoid' function is adopted as the activation function. For both traditional LSTM and AM-LSTM, the number of hidden nodes is also set to 64, with the number of LSTM layers to 2, the number of training epochs to 1000, and the batch size to 32. It is worth noting that due to the inherent randomness in

setting initial solutions and random parameters in these AI techniques, there may be some uncertainty in the experimental results. To comprehensively assess the robustness of the models, we measure and consider the standard deviation based on 20 runs.

**4.1.3. Decomposition and recombination.** Taking the pork price as an example, the price series is broken down by CEEMD to produce intrinsic mode components with various fluctuation scales. These components include six basic mode components, IMF1—IMF6 and one residual component Re. The high-frequency components (IMF1—IMF4) are thought to be the variables responsible to price unpredictability, and the low-frequency components (IMF5—IMF6) exhibit distinct periodic patterns, while the Re component reveals the original price's long-term trend. The decomposed modal components are shown in Fig 5. Decomposing the beef price sequence in the same way yields six IMFs and one residual component, whereas the mutton price sequence also produces seven IMFs and one residual component. Subsequently, the fuzzy entropy of these components can be calculated. For pork, the fuzzy entropy values are sequentially 0.333, 0.2891, 0.2799, 0.2346, 0.1019, 0.0401, and 0.0048; for beef, they are 0.2379, 0.2421, 0.1436, 0.0882, 0.0657, 0.0287, and 0.0153; and for mutton, they are 0.2409, 0.2372, 0.1519, 0.1174, 0.0679, 0.0307, 0.0125, and 0.0081. As the complexity decreases, the fuzzy entropy of each component gradually decreases in order. Finally, the mean fuzzy entropy of each component is computed to establish a threshold for categorizing and rearranging these components. The prices of beef and mutton have reached recombination through the same the decomposition and recombination process. Table 2 presents the recombined results. By reconstructing the original sub-modes into three new parts, the number of sub-prediction models is decreased, leading to an overall enhancement in prediction efficacy.

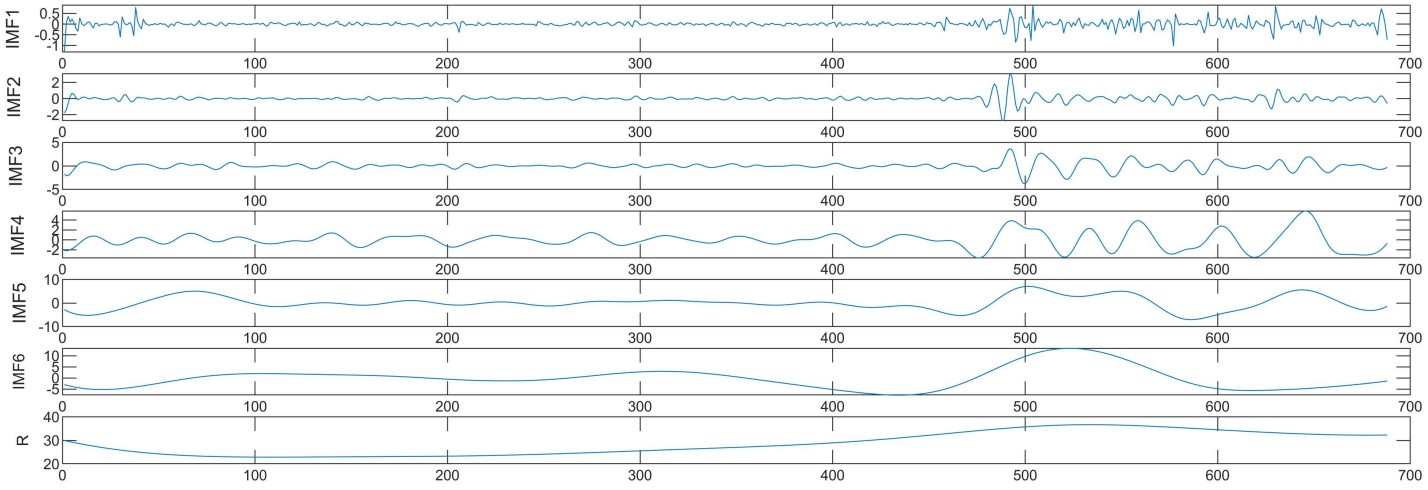

**Fig 5. Decomposition results of pork price.**

**Table 2. Reorganization results of each IMF component.**

| Dataset | Threshold | High PE | Low PE | Trend PE |
|---|---|---|---|---|
| Pork | 0.1833 | IMF1 + IMF2 + IMF3 + IMF4 | IMF5 + IMF6 | Re |
| Beef | 0.1174 | IMF1 + IMF2 + IMF3 | IMF4 + IMF5 + IMF6 | Re |
| Mutton | 0.1083 | IMF1 + IMF2 + IMF3 + IMF4 | IMF5 + IMF6 + IMF7 | Re |

## 4.2. Point predication performance

In point prediction, after completing data decomposition, reconstruction, and characteristics categorization, the product pricing data is directly input into the intelligent prediction algorithm AM-LSTM, without fuzzification processing. The embedding dimension and the prediction step are two crucial factors that must be carefully considered. First, the embedding dimension dictates how much historical data is employed in the model to predict future price changes. Second, the prediction step size specifies how many weeks the model forecast future prices, indicating the out-of-sample predictive ability of the approach [51]. To create a prediction model, the observed price of the m weeks $x_{t-m+1}$, $x_{t-m+2}$, $x_t$ is taken as input sequence, while the actual value of the next n week $x_{t+n}$ serves as the output. m and n are called embedding dimension and prediction step, respectively. By adjusting the two factors m and n to values within the ranges of 1, 2, …, 7, and 1, 2, 3, it is possible to create more realistic and accurate predictive models. Following numerous trials and thorough analysis, we discover that the RMSE and MAPE of pork price prediction are minimized when the prediction step n is set to 1 and the embedding dimension is m = 5. The embedding dimension in the beef and mutton models is still set at 5 for facilitating comparison.

Fig 6 compares the performance of the point prediction step. The CEEMD-FE-AM-LSTM model demonstrates superior stability and prediction accuracy over all. This is because the original time series' complexity and non-stationarity are diminished through the utilization of the CEEMD technique, which fragments it into multiple intrinsic mode functions with unique time scale features. Moreover, comparable complexity components are simultaneously reconstructed using FE into a high-frequency group, a low-frequency group, and a residual group, and AM-LSTM prediction and training are carried out independently, which significantly reduces training time and guarantees high prediction accuracy. Nevertheless, as the prediction step increases, both the MAPE and RMSE undergo substantial increments due to the anomalous fluctuations and volatility int the original price series data, which means that the computational complexity of the prediction learning model rises with the step length.

The suggested model, CEEMD-FE-AM-LSTM, is also compared with other filtered models, such as CEEMD-FE-LSTM, CEEMD-FE-ELM, CEEMD-FE-BP, and unfiltered models, such as LSTM, ELM, and BP, which have different decomposition techniques and prediction algorithms, to validate its effectiveness in point prediction of pork prices. Fig 7 reveals that all eight algorithms effectively track the observed trends in pork price series over time.

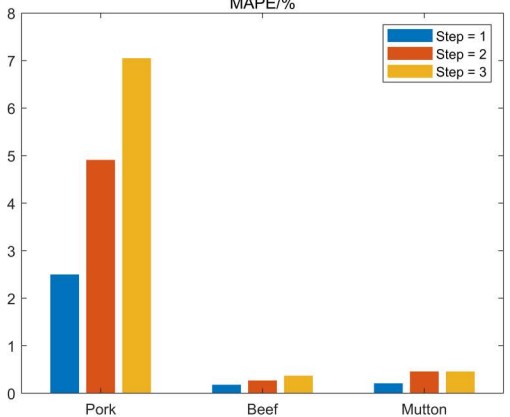 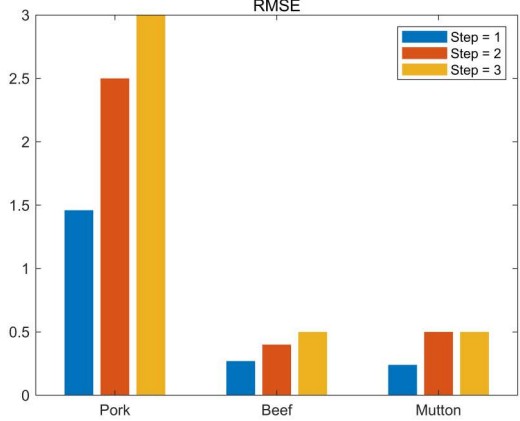

**Fig 6. Point forecasting performance comparison of different prediction steps.**

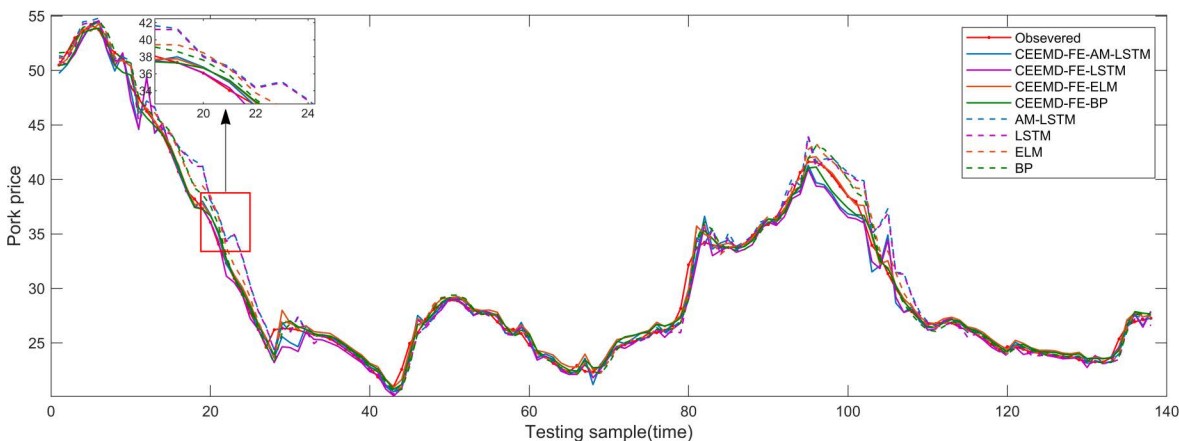

**Fig 7. Point prediction results of different filtered and unfiltered models.**

Additionally, the prediction performance of the CEEMD-FE-AM-LSTM, CEEMD-FE-LSTM, CEEMD-FE-ELM, and CEEMD-FE-BP, which are filtered by CEEMD, FE and AM, are significantly superior to those of the unfiltered ones, i.e., AM-LSTM, LSTM, ELM and BP. This indicates that the proposed prediction system excels in capturing the temporal uncertainty hidden within price data.

## 4.3. Interval prediction performance

For interval prediction, apart from examining the four parameters of embedding dimension, prediction step, decomposition method, and prediction algorithm similar to point prediction, we further discussed the granulation window size (GWS). Choosing a GWS that is ideal for capturing the trend of time series changes can lead to excellent regression results. If the selected window is too lengthy, the resulting fine-grained time series can be too short to build a viable fuzzy rule library. Regression analysis sample size will decrease with a smaller window size, which may make true short-term change detection more difficult and result in results that are more susceptible to noise. So, after reconstructing the modal components, we partition price series that need refining into an equal number of operation windows by different GWS, and then produce upper and lower bounds. With a granulation window size of 3, 4, 5, and an embedding dimension of 5, the AM-LSTM algorithm produced the comparative findings for pork prices in Fig 8. From Fig 8, it evident that at one time point, the actual value exceeds the upper limit of the forecast interval, and at another time point, the actual value falls below the lower limit. The actual values at six different time points, as shown in Fig 9, deviate from the prediction interval, with two points exceeding the upper limit and four falling below the lower limit. Meanwhile, Fig 10 depicts two time points where the actual values fall outside the prediction interval, one point surpassing the upper limit and the other lower than the lower limit. To confirm the relevance and practicality of the approach, research and discussion will also conduct on the pricing of beef and mutton. When GWS = 3, 4, 5, there are 2, 1, 1 points respectively in the beef price prediction that are not within the prediction range, and there are 3, 0, 0 points respectively in the mutton price prediction that are fall outside the predicted range. A good interval prediction model should minimize the interval width without compromising the coverage probability. In this case, where the input sequence is used for the granulation process, FICP represents the probability that the actual price is covered by the predicted interval, while FIAW evaluates the uncertainty

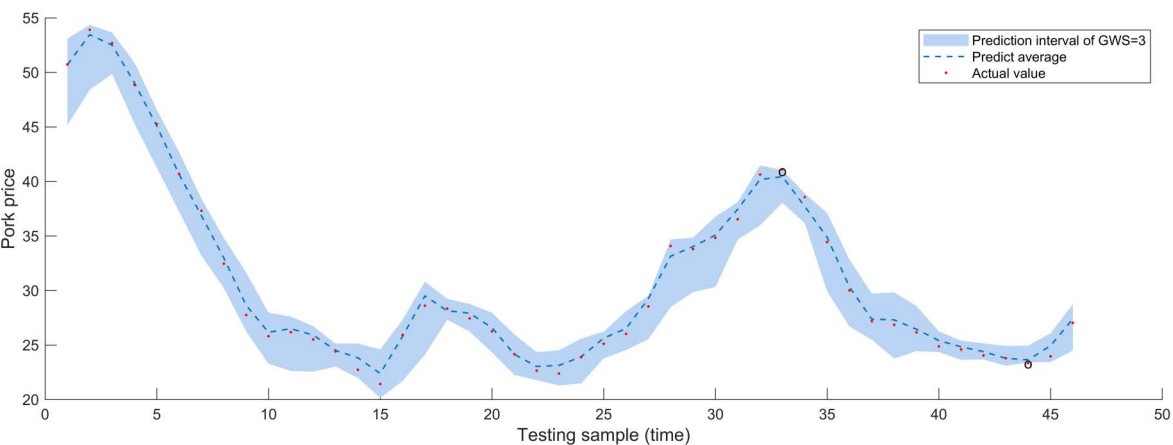

**Fig 8. Impact of granulation window size on interval forecast performance for pork price weeks (GSW = 3).**

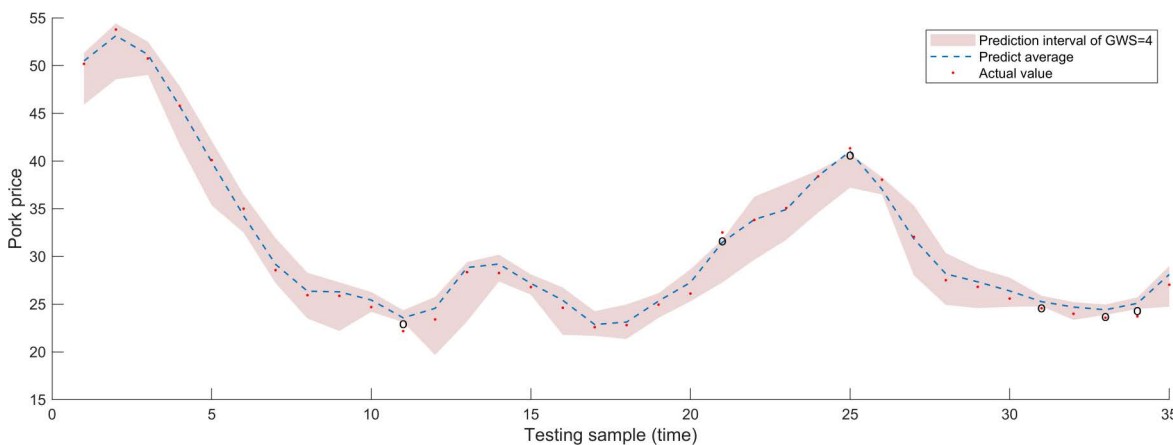

**Fig 9. Impact of granulation window size on interval forecast performance for pork price weeks (GSW = 4).**

information encompassed by the prediction interval. Table 3 provides a comprehensive overview of the price sequence granulation process, highlighting that the interval prediction results for livestock product prices under various GSWs vary significantly. With minimal data points outside the anticipated range during peaks and troughs, the majority of real prices for GWS of three weeks fall inside the predicted range. The prediction effect of the 4- or 5-weeks window size is not as excellent as that of the 3-week window size, but the overall forecast is reasonable, with a high coverage rate and a limited range of projected intervals. As a result, GSW = 3 is chosen for further exploration.

The interval prediction results under various prediction steps are displayed in Fig 11. It can be seen that, depending on the dataset, the system is capable of doing multi-step prediction directly. Unfortunately, as step size increases, so does the prediction error, resulting in a decline in prediction performance consistent with the point prediction analysis results. Taking pork prices as example, the FICP of one-step prediction is 93.481% and the FIAW is 0.135. However, for two-step prediction, the FICP declines by 8.836% and FIAW increases by 0.444%. In the three-step prediction, FICP falls by 25.118% while FIAW rises by 7.032%.

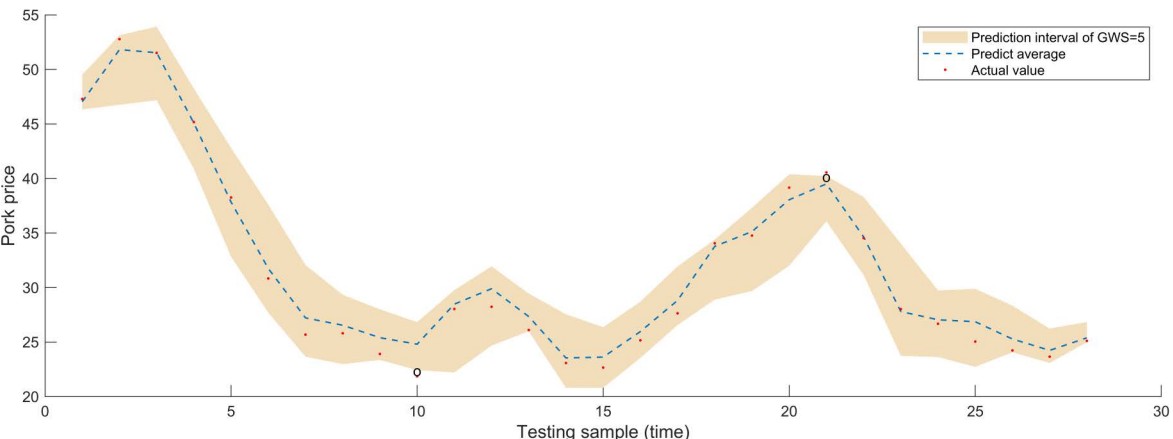

**Fig 10. Impact of granulation window size on interval forecast performance for pork price weeks (GSW = 5).**

**Table 3. Prediction performance under different time intervals.**

| Dataset | Granulation window size | Operation window size | FICP/% | FIAW |
|---|---|---|---|---|
| Pork | **3** | **229** | **93.481** | **0.135** |
| | 4 | 172 | 85.712 | 0.128 |
| | 5 | 138 | 92.861 | 0.205 |
| Beef | **3** | **229** | **97.832** | **0.013** |
| | 4 | 172 | 88.568 | 0.013 |
| | 5 | 138 | 96.428 | 0.022 |
| Mutton | **3** | **229** | **97.828** | **0.017** |
| | 4 | 172 | 97.143 | 0.016 |

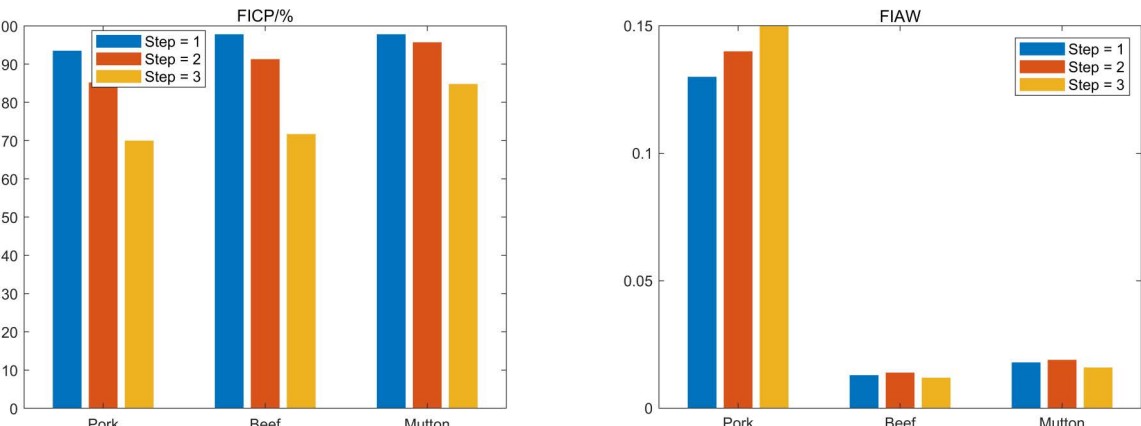

**Fig 11. Interval forecasting performance comparison of different prediction steps.**

We also conduct a comparative analysis between the filtered model and the unfiltered model as presented in point prediction, to further illustrate the superiority of our prediction framework. The parameter selection for these models aligns with that of point prediction.

**Table 4.  Performance comparison of different prediction models.**

| Dataset | Model | MAPE | RMSE | FICP | FIAW | Running time/s |
|---|---|---|---|---|---|---|
| Pork | **CEEMD-FE-FIG-AM-LSTM** | **1.283** | **0.450** | **93.481** | **0.135** | **336.6** |
| | CEEMD-FE-FIG-LSTM | 1.533 | 0.612 | 90.035 | 0.128 | 291.4 |
| | CEEMD-FE-FIG-ELM | 2.458 | 0.771 | 84.782 | 0.112 | 6.7 |
| | CEEMD-FE-FIG-BP | 3.781 | 1.123 | 84.233 | 0.132 | 9.1 |
| | FIG-AM-LSTM | 6.993 | 2.899 | 75,652 | 0.114 | 313.7 |
| | FIG-LSTM | 8.166 | 3.537 | 61.258 | 0.199 | 285.8 |
| | FIG-ELM | 10.702 | 4.678 | 52.614 | 0.333 | 3.8 |
| | FIG-BP | 8.372 | 3.732 | 30.429 | 0.088 | 6.2 |
| Beef | **CEEMD-FE-FIG-AM-LSTM** | **0.153** | **0.173** | **97.832** | **0.013** | **318.7** |
| | CEEMD-FE-FIG-LSTM | 0.588 | 0.626 | 90.184 | 0.013 | 290.2 |
| | CEEMD-FE-FIG-ELM | 1.692 | 1.544 | 64.349 | 0.012 | 7.2 |
| | CEEMD-FE-FIG-BP | 0.392 | 0.402 | 57.829 | 0.011 | 8.1 |
| | FIG-AM-LSTM | 1.674 | 1.574 | 76.683 | 0.009 | 314.6 |
| | FIG-LSTM | 1.832 | 1.780 | 62.732 | 0.031 | 285.5 |
| | FIG-ELM | 2.062 | 1.922 | 59.772 | 0.065 | 4.1 |
| | FIG-BP | 2.164 | 1.089 | 36.124 | 0.007 | 5.0 |
| Mutton | **CEEMD-FE-FIG-AM-LSTM** | **0.183** | **0.186** | **97.828** | **0.017** | **317.3** |
| | CEEMD-FE-FIG-LSTM | 0.188 | 0.193 | 95.284 | 0.020 | 293.2 |
| | CEEMD-FE-FIG-ELM | 0.192 | 0.202 | 93.482 | 0.022 | 6.7 |
| | CEEMD-FE-FIG-BP | 0.812 | 0.791 | 66.960 | 0.020 | 8.2 |
| | FIG-AM-LSTM | 0.654 | 0.674 | 74.353 | 0.010 | 316.5 |
| | FIG-LSTM | 0.582 | 0.612 | 62.564 | 0.012 | 298.3 |
| | FIG-ELM | 0.492 | 0.518 | 56.957 | 0.018 | 3.8 |
| | FIG-BP | 1.158 | 1.164 | 45.222 | 0.011 | 5.3 |

Table 4 shows further insight into the prediction results for pork, beef, and mutton under these eight distinct models. The error between the predicted mean and actual values is measured by the terms MAPE and RMSE, and the uncertainty of the predicted interval is measured by FICP and FIAW. Comparative examination reveals that: (1) In all three scenarios, the filtered model has good predictive performance, with fluctuations roughly consistent with price trends, and most of the measured values fall within the formed prediction range, but about 75% of the observations fall outside of the prediction interval for unfiltered models, which demonstrates the advantages of CEEMD and FE in extracting dynamic uncertainty. (2) the CEEMD-FE-FIG-AM-LSTM model presents in this study achieves narrow interval width and high interval coverage probability. Mutton price, for example, the CEEMD-FE-FIG-AM-LSTM model obtains a coverage probability of 97.828%, the CEEMD-FE-FIG-LSTM model obtains a coverage probability of 95.284%, the CEEMD-FE-FIG-ELM model reaches a coverage probability of 93.482%, and the CEEMD-FE-FIG-BP model attains a coverage rate of 66.960%. Notably, the interval width for all three models is approximately 0.02. Which due to the benefits of AM-LSTM, LSTM excels at processing long sequence data, effectively alleviating gradient vanishing problems and capturing long-term dependencies. The introduction of attention mechanism enables LSTM to dynamically focus on key information in the sequence, improving its ability to capture important features and helping the model maintain memory of long-distance dependencies. However, more complicated learning procedures and parameter adjustments are needed for BP, which occasionally impact their predicted accuracy. (3) From the perspective of runtime, models that have been filtered

by CEEMD-FE exhibit a notable improvement in prediction accuracy despite incurring only a slight increase in computational cost compared to those without filtering. In particular, the CEEMD-FE-FIG-AM-LSTM model proposed in this paper has demonstrated a marked advantage in predictive performance across three datasets, boasting the highest FICP and the narrowest FIAW. Although this model has a relatively long runtime, it is still within an acceptable range. Specifically, in pork price prediction, the runtime of the CEEMD-FE-FIG-AM-LSTM model is 336.6 seconds, which is slightly more than the FIG-AM-LSTM (313.7 seconds) and FIG-LSTM (285.8 seconds) models that do not utilize CEEMD. However, the FICP of this model reaches as high as 93.481%, and its MAPE value is only 1.283, significantly lower than the latter two models of 6.993 and 8.166, respectively, fully demonstrating its superior prediction accuracy. In beef and mutton price prediction, the runtimes of the model are 318.7 seconds and 317.3 seconds, respectively, with minimal differences compared to models without CEEMD. But its MAPE values are lower, at 0.153 and 0.183, further validating its excellent prediction performance. In contrast, while the CEEMD-FE-FIG-ELM model has a significant advantage in runtime, with operation times of 6.7 seconds, 7.2 seconds, and 6.7 seconds for the three datasets, its prediction accuracy is relatively lower, and its FICP values are also smaller. Therefore, in practical applications, we need to strike a balance between speed and accuracy based on specific requirements. For scenarios with high real-time performance, such as real-time traffic flow prediction, the ELM model is more suitable due to its operational efficiency. Nevertheless, in situations demanding extreme prediction accuracy, such as long-term investment decisions in financial markets, electricity demand forecasting in the energy industry, or crop yield and price prediction in agriculture, the AM-LSTM model is more competitive due to its higher prediction accuracy. In conclusion, applying this approach can significantly enhance the prediction performance of livestock product prices, offering valuable technical assistance for efficient livestock product price monitoring.

At present, the prevailing methodology for interval prediction is to construct confidence intervals by combining point prediction results with the probabilistic properties of the original data. Consequently, this comparison method is chosen as a benchmark for analysis in the study. The process commenced with the utilization of a spectrum of prevalent distribution functions, encompassing the normal, Weibull, lognormal, exponential, and gamma distributions, to model the raw data. Subsequently, interval predictions are executed at varying confidence levels (a = 99%, a = 95%, a = 90%), leveraging the fitted distribution functions in conjunction with point prediction results. The results of these predictions are meticulously detailed in Table 5. An optimal interval prediction model aims for a lower FIAW value (i.e., a smaller prediction interval width) and a higher FICP value (i.e., the proportion of true values contained in the prediction interval). Our analysis reveals that as the confidence level decreases from 99% to 90%, the predictive certainty for these quintessential models decreases when the data aligns with their respective distributional assumptions, while the predictive accuracy increases. In terms of pork price prediction, the normal distribution has a slightly higher FIAW (e.g., 1.611), indicating a broad prediction interval and, as a result, difficulty in defining an appropriate pricing range, even while it retains some coverage proficiency at high confidence levels (e.g., FICP of 97.102 at 99%). The Weibull distribution's prediction accuracy is limited since it lacks a small FIAW, even though it achieves a FICP of 100 at a high confidence level. The lognormal distribution also struggles with a bigger FIAW at high confidence levels, resulting in an imprecise control over the price range. The exponential distribution typically exhibits a high FIAW during a range of confidence levels, with a peak FIAW of 2.024 at 99% confidence level, leading to an excessively broad spectrum of predicted prices, which is detrimental to practical applications. In both FICP and FIAW, the gamma distribution performs excellently, although it has not yet reached an optimal equilibrium state. On the other hand,

**Table 5. Comparison results of the proposed model and distribution-based interval prediction methods.**

|  |  | Pork | | Beef | | Mutton | |
|---|---|---|---|---|---|---|---|
| Level | Distribution | FICP(%) | FIAW | FICP(%) | FIAW | FICP(%) | FIAW |
| a = 99% | Normal | 97.102 | 1.611 | 100 | 0.823 | 100 | 0.744 |
|  | Weibull | 100 | 1.414 | 100 | 0.782 | 97.101 | 0.711 |
|  | Lognormal | 100 | 1.569 | 100 | 0.823 | 100 | 0.744 |
|  | Exponential | 100 | 2.024 | 100 | 1.669 | 100 | 1.734 |
|  | Gamma | 100 | 1.379 | 100 | 0.835 | 100 | 0.754 |
| a = 95% | Normal | 92.033 | 1.226 | 55.072 | 0.626 | 50.000 | 0.566 |
|  | Weibull | 93.484 | 1.064 | 9.424 | 0.611 | 19.574 | 0.558 |
|  | Lognormal | 93.482 | 1.033 | 55.074 | 0.626 | 50.000 | 0.566 |
|  | Exponential | 100 | 1.401 | 100 | 1.155 | 100 | 1.200 |
|  | Gamma | 92.751 | 1.007 | 76.088 | 0.635 | 63.771 | 0.574 |
| a = 90% | Normal | 89.862 | 1.029 | 7.245 | 0.526 | 7.246 | 0.475 |
|  | Weibull | 90.579 | 0.891 | 1.447 | 0.517 | 3.623 | 0.474 |
|  | Lognormal | 89.864 | 0.816 | 7.246 | 0.526 | 7.246 | 0.475 |
|  | Exponential | 94.926 | 1.126 | 100 | 0.928 | 100 | 0.965 |
|  | Gamma | 89.861 | 0.831 | 7.972 | 0.533 | 7.973 | 0.481 |
| **The proposed** |  | **93.481** | **0.135** | **97.832** | **0.013** | **97.828** | **0.017** |

the CEEMD-FE-FIG-AM-LSTM model proposed in this study has proven to have a number of benefits. With an amazing FICP value of 93.481, the model is able to approach the true value of agricultural product prices within the forecast interval, guaranteeing the prediction's trustworthiness. Meanwhile, the FIAW value is just 0.135, which is significantly less than the FIAW values of all the previously listed common distributions at different levels of confidence, signifying that the model can furnish more precise prediction intervals. Without significantly compromising the coverage of real parameters, this model provides more accurate and useful information for real-world applications by substantially decreasing the prediction range. Similar conclusions can be drawn from the beef and mutton datasets. Furthermore, it is pertinent to underscore that, despite their operational simplicity, the previously mentioned traditional interval prediction techniques do have some significant drawbacks. In particular, a common problem is the difficulty of precisely capturing the empirical distributional subtleties of prices data, as demonstrated by commodities like pork, beef, and mutton. These models' application and effectiveness in the predictive analytics field are significantly limited by their inability to properly reconcile with the inherent distributional characteristics of the data. The proposed interval prediction model, conversely, demonstrates a notable advantage in its ability to handle the intricacies of the data. It offers a more dependable and advanced prediction framework that can provide interval predictions that are both narrow and highly inclusive of the true parameters. As a result, the contributions of this study highlight the transformative potential of the suggested method within the larger framework of economic forecasting and decision-making, while also advancing the theoretical understanding of interval prediction and holding promising implications for practical applications.

## 4.4. The relationship between point prediction and interval prediction

Fig 8 shows that the height of the interval forecast average value, displayed in the blue dashed line, is essentially comparable with the real pork price, indicated by the red dots, both have an RMSE of 0.450 and a MAPE of 1.283% in CEEMD-FE-FIG-AM-LSTM. On the other hand, the unfiltered model FIG-AM-LSTM causes a small variation in the height of the two interval predictions, but the values—which have 2.899 RMSE and 6.992% MAPE—remain extremely

close. This conclusion has also been validated in the other three comparative models. Consequently, it makes sense for us to utilize the interval prediction's average value as the point prediction result. Additionally, a comparison of interval prediction techniques in the literature [36,37] reveals that the method proposed in this paper is also versatile and can be applied to a range of time series, including daily, weekly data, and monthly data, and does not have any particular criteria for data sources. Furthermore, it is capable of directly predicting intervals without relying on computations from statistical theory or prior distribution assumptions. Finally, our method can substitute point prediction to yield more accurate prediction results in both fuzzy granulations mean sequence prediction and interval range prediction. To sum up, the interval prediction system that has been shown has robust predictive capabilities and consistency for both interval and point prediction. Whereas interval prediction depicts future trend and range of product prices, point prediction delivers average value of the price series in future.

## 5. Conclusion

To capture evolving patterns and forecast future trend, it is crucial from a practical standpoint to conduct uncertainty analysis of the livestock products price with noise and non-stationary features. The paper offers an effective method for the uncertainty analysis of price series based on fuzzy mathematics and traditional machine learning fully integrating the advantages of CEEMD, FE, FIG, and AM-LSTM algorithms to examine the tendency and range of product price changes. The deconstructed sequence is first reconstructed into high-frequency, low-frequency, and residual groups with feature entropy judgment by integrating CEEMD and FE in order to lessen data volatility and complexity. Then, the restructured pricing data is divided into many information granules according to FIG, which makes it possible to retrieve relevant information about the maximum, minimum, and mean. Subsequently, AM-LSTM is applied to forecast the granulated data and linearly integrates the projected values to determine the final prediction interval. Finally, Using the comprehensive evaluation criteria, comparison and discussion with the other three filtering models and four non-filtering models to validate the model's accuracy and efficacy in point and interval prediction. Based on the comprehensive evaluation criteria, the proposed model was systematically applied to three datasets: pork, beef, and mutton. By comparing and analyzing with other filtering and non-filtering models, the accuracy and effectiveness of this model in point and interval prediction have been verified. The following conclusions can be made:

(1) The CEEMD decomposition method is used to decompose product prices, which reduces the mutual interference of information at different time scales. On this basis, the FE is introduced to assess each component's complexity, and reassemble the original components into new groups, which ensures that the method can fully capture both the linear and nonlinear characteristics of pricing data, while also avoiding high computational complexity associated by redundant components. This processing technique provides a strong basis for achieving accurate interval prediction results.

(2) For the first time, the concept of FIG is utilized to successfully analyze the uncertainty of price data by mining product prices and extracting necessary effective information. And based on LSTM with attention mechanism, interval value data is used to represent the inherent inaccuracy and fuzziness of prices. This innovative method is of great significance for early risk warning of livestock product prices.

(3) Our proposed prediction model achieves dual capabilities of point prediction and interval prediction. In addition to traditional RMSE and MAPE measurement techniques, FICP

and FIAW have been innovatively introduced to more comprehensively and meticulously evaluate the performance of the model in point prediction and interval prediction. These works not only deepen our understanding of model performance and influencing variables, but also significantly optimize the performance of deep learning models in data prediction.

According to the investigation and discussion, future research can expand around the following aspects: First, various metrics to evaluate prediction intervals and novel prediction algorithms can be chosen for comparison research. Secondly, apart from to livestock product prices, the proposed prediction model can also be extended to interval-valued time series prediction tasks in other fields, such as high volatility financial and energy markets, to fully verify its universality and universality. Finally, the theoretical background of the proposed prediction system can be further explored, allowing for the determination of the confidence level of the prediction interval generated by the prediction system and compared with the prediction interval derived from statistical distribution theory.

## Author contributions

**Conceptualization:** weimin ma, lingling peng, hu Chen.

**Data curation:** hu Chen.

**Formal analysis:** lingling peng, haisheng yan.

**Methodology:** lingling peng.

**Software:** lingling peng.

**Supervision:** weimin ma.

**Writing – original draft:** lingling peng.

**Writing – review & editing:** weimin ma, haisheng yan.

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
