## [Decision Letter · Decision Letter 0]

1 Nov 2024

PONE-D-24-41802

Interval price prediction of livestock product based on fuzzy mathematics and improved LSTM

PLOS ONE

Dear Dr. peng,

Thank you for submitting your manuscript to PLOS ONE. After careful consideration, we feel that it has merit but does not fully meet PLOS ONE’s publication criteria as it currently stands. Therefore, we invite you to submit a revised version of the manuscript that addresses the points raised during the review process.

We look forward to receiving your revised manuscript.

Kind regards,

Jinran Wu, PhD

Academic Editor

PLOS ONE

Journal Requirements:

4. Please note that your Data Availability Statement is currently missing the repository name and/or the DOI/accession number of each dataset OR a direct link to access each database. If your manuscript is accepted for publication, you will be asked to provide these details on a very short timeline. We therefore suggest that you provide this information now, though we will not hold up the peer review process if you are unable.

5. Please remove your figures from within your manuscript file, leaving only the individual TIFF/EPS image files, uploaded separately. These will be automatically included in the reviewers’ PDF**.**

Reviewers' comments:

Reviewer's Responses to Questions

**Comments to the Author**

1. Is the manuscript technically sound, and do the data support the conclusions?

Reviewer #1: Yes

2. Has the statistical analysis been performed appropriately and rigorously? 

Reviewer #1: Yes

3. Have the authors made all data underlying the findings in their manuscript fully available?

Reviewer #1: Yes

4. Is the manuscript presented in an intelligible fashion and written in standard English?

Reviewer #1: Yes

5. Review Comments to the Author

Reviewer #1: 1 Please clarify the third equation for the meaning of the percentage symbol that appears.

2 How does the accuracy of this interval prediction method compare to other interval prediction methods? It is recommended that the authors conduct an experimental comparison with other interval prediction models to validate the effectiveness of their approach.

3 It is suggested that the authors provide a more detailed description of how the prediction intervals are generated using FIG. If possible, including a flowchart to illustrate this process would greatly enhance understanding.

6. PLOS authors have the option to publish the peer review history of their article (what does this mean? ). If published, this will include your full peer review and any attached files.

**Do you want your identity to be public for this peer review?** For information about this choice, including consent withdrawal, please see our Privacy Policy .

Reviewer #1: No

---

## [Author Response · Author response to Decision Letter 1]

3 Dec 2024

Journal Requirements:

1. Please ensure that your manuscript meets PLOS ONE's style requirements, including those for file naming. The PLOS ONE style templates can be found at https://journals.plos.org/plosone/s/file?id=wjVg/

PLOSOne_formatting_sample_main_body.pdf and https://journals.plos.or

g/plosone/s/file?id=ba62/PLOSOne_formatting_sample_title_authors_affiliations.pdf

Response:

Thanks very much for the following valuable comments, we have carefully revised the manuscript according to PLOS ONE's style requirements, with the detailed point-to-point responses listed as follows.

Response:

Thanks very much for this helpful suggestion, we have shared our code in protocols.io to enhance the reproducibility of results.

DOI: dx.doi.org/10.17504/protocols.io.x54v9rj2pv3e/v1(Private link for reviewers: https://www.protocols.io/private/7AA710E9AD4811EF9F120A58A9FEAC02 to be removed before publication.)

Response:

Thanks for the careful examination. We have removed the funding-related text from the manuscript.

4. Please note that your Data Availability Statement is currently missing the repository name and/or the DOI/accession number of each dataset OR a direct link to access each database. If your manuscript is accepted for publication, you will be asked to provide these details on a very short timeline. We therefore suggest that you provide this information now, though we will not hold up the peer review process if you are unable.

Response:

Thanks very much for the careful examination. We have provided a direct link to access each database (https://www.ceicdata.com.cn/) and added the DOI numbers for each dataset (https://doi.org/10.3886/E211561V1).

Based on weekly retail price data of agricultural products monitored from 500 markets across China by the Ministry of Agriculture and Rural Affairs, as sourced from CEIC Database (https://www.ceicdata.com.cn/) and illustrated in Figure 4, the prices of major livestock products in China, encompassing pork, beef, and mutton, have exhibited a notable upward trajectory since the year 2009.

Please see the revisions in the first sentence of Section 4.1.

5. Please remove your figures from within your manuscript file, leaving only the individual TIFF/EPS image files, uploaded separately. These will be automatically included in the reviewers’ PDF.

Response:

Thanks for the careful examination. Accordingly, we have removed all figures from within my manuscript file.

Reviewers' comments:

Reviewer's Responses to Questions

Comments to the Author

1. Is the manuscript technically sound, and do the data support the conclusions?

Reviewer #1: Yes

2. Has the statistical analysis been performed appropriately and rigorously?

Reviewer #1: Yes

3. Have the authors made all data underlying the findings in their manuscript fully available?

Reviewer #1: Yes

4. Is the manuscript presented in an intelligible fashion and written in standard English?

Reviewer #1: Yes

5. Review Comments to the Author

Reviewer #1:

1 Please clarify the third equation for the meaning of the percentage symbol that appears.

Response:

Thanks for the careful examination. We have revised these minor questions in the revised paper.

Where, xt and ht represent the input and output information of the hidden layer. C is the memory unit, C(~) is a candidate unit that ultimately determines which information will be added to C through the control of input gates and forget gates. W is the weight parameter matrix, and b is the bias. σ(.) and β(.) are the sigmoid and tanh activation functions. Attention mechanism can be introduced to improve the LSTM structure and enhance feature extraction efficiency.

Please see the first paragraph and Formula 3 in Section 2.4.

2 How does the accuracy of this interval prediction method compare to other interval prediction methods? It is recommended that the authors conduct an experimental comparison with other interval prediction models to validate the effectiveness of their approach.

Response:

According to the valuable comment, we have added comparative experiments with other interval prediction models, to validate the effectiveness of our proposed method.

At present, the prevailing methodology for interval prediction is to construct confidence intervals by combining point prediction results with the probabilistic properties of the original data. Consequently, this comparison method is chosen as a benchmark for analysis in the study. The process commenced with the utilization of a spectrum of prevalent distribution functions, encompassing the normal, Weibull, lognormal, exponential, and gamma distributions, to model the raw data. Subsequently, interval predictions are executed at varying confidence levels (a=99%, a=95%, a=90%), leveraging the fitted distribution functions in conjunction with point prediction results. The results of these predictions are meticulously detailed in Table 5. An optimal interval prediction model aims for a lower FIAW value (i.e., a smaller prediction interval width) and a higher FICP value (i.e., the proportion of true values contained in the prediction interval). Our analysis reveals that as the confidence level decreases from 99% to 90%, the predictive certainty for these quintessential models decreases when the data aligns with their respective distributional assumptions, while the predictive accuracy increases. In terms of pork price prediction, the normal distribution has a slightly higher FIAW (e.g., 1.611), indicating a broad prediction interval and, as a result, difficulty in defining an appropriate pricing range, even while it retains some coverage proficiency at high confidence levels (e.g., FICP of 97.102 at 99%). The Weibull distribution's prediction accuracy is limited since it lacks a small FIAW, even though it achieves a FICP of 100 at a high confidence level. The lognormal distribution also struggles with a bigger FIAW at high confidence levels, resulting in an imprecise control over the price range. The exponential distribution typically exhibits a high FIAW during a range of confidence levels, with a peak FIAW of 2.024 at 99% confidence level, leading to an excessively broad spectrum of predicted prices, which is detrimental to practical applications. In both FICP and FIAW, the gamma distribution performs excellently, although it has not yet reached an optimal equilibrium state. On the other hand, the CEEMD-FE-FIG-AM-LSTM model proposed in this study has proven to have a number of benefits. With an amazing FICP value of 93.481, the model is able to approach the true value of agricultural product prices within the forecast interval, guaranteeing the prediction's trustworthiness. Meanwhile, the FIAW value is just 0.135, which is significantly less than the FIAW values of all the previously listed common distributions at different levels of confidence, signifying that the model can furnish more precise prediction intervals. Without significantly compromising the coverage of real parameters, this model provides more accurate and useful information for real-world applications by substantially decreasing the prediction range. Similar conclusions can be drawn from the beef and mutton datasets. Furthermore, it is pertinent to underscore that, despite their operational simplicity, the previously mentioned traditional interval prediction techniques do have some significant drawbacks. In particular, a common problem is the difficulty of precisely capturing the empirical distributional subtleties of prices data, as demonstrated by commodities like pork, beef, and mutton. These models' application and effectiveness in the predictive analytics field are significantly limited by their inability to properly reconcile with the inherent distributional characteristics of the data. The proposed interval prediction model, conversely, demonstrates a notable advantage in its ability to handle the intricacies of the data. It offers a more dependable and advanced prediction framework that can provide interval predictions that are both narrow and highly inclusive of the true parameters. As a result, the contributions of this study highlight the transformative potential of the suggested method within the larger framework of economic forecasting and decision-making, while also advancing the theoretical understanding of interval prediction and holding promising implications for practical applications.

Please see the added discussion and Table 5 in the last paragraph of Section 4.4.

3 It is suggested that the authors provide a more detailed description of how the prediction intervals are generated using FIG. If possible, including a flowchart to illustrate this process would greatly enhance understanding.

Response:

Thanks for this valuable suggestion. Accordingly, we have added a discussion on the detailed steps of the entire prediction model and visually demonstrated the generation process of the prediction interval.

The hybrid model described in this work not only predicts the future average product value as a point, but it also predicts the future price fluctuation range as an interval. In light of the benefits of empirical modal decomposition in series smoothing, the advantages of fuzzy information granulation in the granulation interval, and the superior performance of long and short-term memory neural networks in time series data prediction, this paper selects the CEEMD decomposition model, FIG granulation model and AM-LSTM network prediction model, and also adopts the FE algorithm to recombination the components after CEEMD decomposition. Finally, we propose a combined CEEMD-FE-FIG-AM-LSTM prediction model. Fig 3 illustrates the specific steps in this procedure.

(1) Data composition. The initial price sequence X = {x1, x2, … , xn}, with a length of N, can be efficiently decomposed into serval IMF components and one residual component by CEEMD. These components capture the dynamic characteristics of the price series at different scales.

(2) Group allocation. The average fuzzy entropy of each IMF component is computed, serving as a criterion to assess the temporal complexity of the series. Based on this criterion, the IMF components are categorized into three groups: the first group includes components with fuzzy entropy higher than a specified threshold, the second group consists of components with fuzzy entropy lower than the threshold, and the third group encompasses the remaining components. By reducing the dimensionality of IMF components, we aim to simplify the construction complexity of the predictive model.

(3) Characteristics categorization. The FIG method is utilized to divide the three reconstructed modal component groups into non-overlapping, fixed-width subsequences. Each subsequence represents a time window, denoted as w1, w2, … , wt. Within each time window, effective information is extracted using a triangular membership function and represented by fuzzy particles G1, G2, … , Gt. These fuzzy particles form three sets of sequences: {a1, a2, … , at}, {m1, m2, … , mt} and {b1, b2, … , bt}, representing low (LOW), range (R), and upper bounds (UP) respectively. These sequences reflect the average and dispersion range of price changes, where LOW and UP serve as the lower and upper bounds for interval prediction, and R represents the overall trend of price changes. To explore the influence of time intervals on the prediction system, various time windows will be selected for granulation in subsequent experiments.

(4) AM-LSTM prediction. AM-LSTM model is employed to train and predict for the three modal component groups separately. By incorporating the attention mechanism, the AM-LSTM model can better capture long-term dependencies in the time series data, thereby enhancing prediction accuracy.

(5) Forecast results superposition. For the prediction results of the three modal component groups, the final interval prediction result is obtained by linearly aggregating the LOW and UP sequences of each group, while the aggregation result of the R sequences serves as the final point prediction result. This aggregation strategy not only provides interval predictions for price changes but also offers more precise market trend analysis through point prediction results.

Please see the revisions in the first paragraph of Section 3 and Figure 3.

6. PLOS authors have the option to publish the peer review history of their article (what does this mean?). If published, this will include your full peer review and any attached files.

Do you want your identity to be public for this peer review? For information about this choice, including consent withdrawal, please see our Privacy Policy.

Reviewer #1: No

While revising your submission, please upload your figure files to the Preflight Analysis and Conversion Engine (PACE) digital diagnostic tool, https://pacev2.apexcovantage.com/. PACE helps ensure that figures meet PLOS requirements. To use PACE, you must first register as a user. Registration is free. Then, login and navigate to the UPLOAD tab, where you will find detailed instructions on how to use the tool. If you encounter any issues or have any questions when using PACE, please email PLOS at figures@plos.org . Please note that Supporting Information files do not need this step.

In compliance with data protection regulations, you may request that we remove your personal registration details at any time. (Remove my information/

---

## [Decision Letter · Decision Letter 1]

27 Dec 2024

PONE-D-24-41802R1Interval price prediction of livestock product based on fuzzy mathematics and improved LSTMPLOS ONE

Dear Dr. peng,

Thank you for submitting your manuscript to PLOS ONE. After careful consideration, we feel that it has merit but does not fully meet PLOS ONE’s publication criteria as it currently stands. Therefore, we invite you to submit a revised version of the manuscript that addresses the points raised during the review process.

We look forward to receiving your revised manuscript.

Kind regards,

Jinran Wu, PhD

Academic Editor

PLOS ONE

Reviewers' comments:

Reviewer's Responses to Questions

**Comments to the Author**

1. If the authors have adequately addressed your comments raised in a previous round of review and you feel that this manuscript is now acceptable for publication, you may indicate that here to bypass the “Comments to the Author” section, enter your conflict of interest statement in the “Confidential to Editor” section, and submit your "Accept" recommendation.

Reviewer #2: (No Response)

Reviewer #3: All comments have been addressed

2. Is the manuscript technically sound, and do the data support the conclusions?

Reviewer #2: Yes

Reviewer #3: Yes

3. Has the statistical analysis been performed appropriately and rigorously? 

Reviewer #2: Yes

Reviewer #3: Yes

4. Have the authors made all data underlying the findings in their manuscript fully available?

Reviewer #2: Yes

Reviewer #3: Yes

5. Is the manuscript presented in an intelligible fashion and written in standard English?

Reviewer #2: No

Reviewer #3: Yes

6. Review Comments to the Author

Reviewer #2: A comprehensive livestock price prediction model with joint point and interval prediction capabilities is proposed in the work. An empirical study was conducted on the weekly price data of pork, beef, and mutton in China from 2009 to 2023, incorporating discussions on different embedding dimensions, prediction step, fuzzy granulation window sizes, decomposition techniques, and prediction algorithms. Some revisions are needed.

1.What is the number of layers of CEEMD decomposition? How does the author make sure this is optimal?

2.The components are divided into three groups according to the fuzzy entropy value. The author does not make it clear here that fuzzy entropy is to be used instead of some other statistic. Why can't other statistics be used?

3.AM-LSTM is not a very new algorithm. Both AM and LSTM have been used in several different research efforts. And their integration is very common.

4.Many integrated methods for RUL prediction are not discussed in the work. Such as VMD, PF and GPR are integrated to the battery RUL prediction. BLS and LSTM integrated to the battery RUL prediction. PSO, RVM and voltage for analog circuit fault prognostics, etc.

5.Parameter configuration, parameter adjustment scheme is very important. The author needs to give. In particular, the parameter configuration scheme of the comparison method also needs to be given.

6.The author needs to give the calculation time of the proposed method and the calculation time of the comparison method, so as to verify the effectiveness of the proposed method.

Reviewer #3: Suggest the author refer to the following literature to improve the research background and current status of the algorithm studied in this article.

[1]R. B. Kagade and N. Vijayaraj, “Intrusion detection via optimal tuned LSTM model with trust and risk level evaluation,” International Journal of Bio-Inspired Computation, vol. 23, no. 1, pp. 39–52, Jan. 2024.

[2]J. Tang, W. Gu, Z. Lei, and S. Gao, “A decode-based chaotic adaptive differential evolution for fuzzy job-shop scheduling problem,” International Journal of Bio-Inspired Computation, vol. 24, no. 4, pp. 212–222, Jan. 2024.

7. PLOS authors have the option to publish the peer review history of their article (what does this mean? ). If published, this will include your full peer review and any attached files.

**Do you want your identity to be public for this peer review?** For information about this choice, including consent withdrawal, please see our Privacy Policy .

Reviewer #2: No

Reviewer #3: No

---

## [Author Response · Author response to Decision Letter 2]

20 Jan 2025

Reviewers' comments:

Reviewer's Responses to Questions

Comments to the Author

1. If the authors have adequately addressed your comments raised in a previous round of review and you feel that this manuscript is now acceptable for publication, you may indicate that here to bypass the “Comments to the Author” section, enter your conflict of interest statement in the “Confidential to Editor” section, and submit your "Accept" recommendation.

Reviewer #2: (No Response)

Reviewer #3: All comments have been addressed

2. Is the manuscript technically sound, and do the data support the conclusions?

Reviewer #2: Yes

Reviewer #3: Yes

3. Has the statistical analysis been performed appropriately and rigorously?

Reviewer #2: Yes

Reviewer #3: Yes

4. Have the authors made all data underlying the findings in their manuscript fully available?

Reviewer #2: Yes

Reviewer #3: Yes

5. Is the manuscript presented in an intelligible fashion and written in standard English?

Reviewer #2: No

Reviewer #3: Yes

6. Review Comments to the Author

Reviewer #2: A comprehensive livestock price prediction model with joint point and interval prediction capabilities is proposed in the work. An empirical study was conducted on the weekly price data of pork, beef, and mutton in China from 2009 to 2023, incorporating discussions on different embedding dimensions, prediction step, fuzzy granulation window sizes, decomposition techniques, and prediction algorithms. Some revisions are needed.

1. What is the number of layers of CEEMD decomposition? How does the author make sure this is optimal?

Response:

Thanks very much for the valuable comment. We have added the detailed steps of CEEMD decomposition in Section 2.1 and a discussion on the results in Section 4.1.3 to validate its effectiveness.

The decomposition process of CEEMD can be expressed as:

(1)

Where x(t) represents the original signal, Ci(t) is the i-th IMF component, and rn(t) stands for the residual component. In the CEEMD, the original signal undergoes a preprocessing step where pairs of positive and negative Gaussian white noise, denoted by u+ and u-, respectively, are added to generate positive and negative sequences.

(2)

Subsequently, EMD decomposition is applied to each of these noise-added sequences, corresponding IMF components and residual components are obtained. The final result of the CEEMD decomposition is as follows:

(3)

By adopting this method, CEEMD successfully minimizes noise interference during decomposition and enhances the physical significance and interpretability of the IMFs. This study sets the chosen iteration number M to 200 and the white noise amplitude k to 0.2 based on the features of the research data.

Please see the revisions in Section 2.1.

Taking the pork price as an example, the price series is broken down by CEEMD to produce intrinsic mode components with various fluctuation scales. These components include six basic mode components, IMF1—IMF6 and one residual component Re. The high-frequency components (IMF1—IMF4) are thought to be the variables responsible to price unpredictability, and the low-frequency components (IMF5—IMF6) exhibit distinct periodic patterns, while the Re component reveals the original price’s long-term trend. The decomposed modal components are shown in Fig 5. Decomposing the beef price sequence in the same way yields six IMFs and one residual component, whereas the mutton price sequence also produces seven IMFs and one residual component.

Please see the revisions in Section 4.1.3.

After CEEMD decomposition, the price data for pork, beef, and mutton are broken down into 7, 7, and 8 layers, respectively, as shown in Fig. 1. CEEMD is an advanced decomposition method that automatically determines the number of layers based on the signal's intrinsic properties and frequency components, ensuring each IMF captures a specific frequency component. The number of layers is determined adaptively without manual intervention. To optimize the CEEMD decomposition, we:

1) Preprocess the data by denoising and handling missing values.

2) Determine the optimal noise parameters through experiments (iteration

number 200 and the white noise amplitude 0.2).

3) Verify each IMF layer's compliance with the defining conditions and

check the reconstruction accuracy.

4) Compare the CEEMD method with non-decomposition approaches (as

detailed in Table 4 of the manuscript), confirming its effectiveness in separating signal components and providing reliable features for prediction.

In summary, we have ensured the optimality of the CEEMD decomposition layers, which lays a solid foundation for accurate price prediction.

Fig 1. Decomposition results of pork, beef and mutton prices.

2. The components are divided into three groups according to the fuzzy entropy value. The author does not make it clear here that fuzzy entropy is to be used instead of some other statistic. Why can't other statistics be used?

Response:

Thanks for this insightful comment. We have added a detailed explanation of the fuzzy entropy analysis performed on the decomposed sequences of pork, beef, and mutton prices in Section 4.1.3.

Subsequently, the fuzzy entropy of these components can be calculated. For pork, the fuzzy entropy values are sequentially 0.333, 0.2891, 0.2799, 0.2346, 0.1019, 0.0401, and 0.0048; for beef, they are 0.2379, 0.2421, 0.1436, 0.0882, 0.0657, 0.0287, and 0.0153; and for mutton, they are 0.2409, 0.2372, 0.1519, 0.1174, 0.0679, 0.0307, 0.0125, and 0.0081. As the complexity decreases, the fuzzy entropy of each component gradually decreases in order. Finally, the mean fuzzy entropy of each component is computed to establish a threshold for categorizing and rearranging these components. The prices of beef and mutton have reached recombination through the same the decomposition and recombination process. Table 2 presents the recombined results. By reconstructing the original sub-modes into three new parts, the number of sub-prediction models is decreased, leading to an overall enhancement in prediction efficacy.

Please see the revisions in Section 4.1.3.

We chose fuzzy entropy over other statistical measures to group the IMFs decomposed by CEEMD for these reasons:

1) Advantages of Fuzzy Entropy. It uses fuzzy membership functions,

making the entropy value sensitive to parameter changes. This helps accurately capture the complexity of IMFs related to price series fluctuations, providing a solid basis for grouping.

2) Robustness to noise. Fuzzy entropy reduces noise interference through mean calculations and fuzzy functions. This allows for clear identification of each IMF's true characteristics, reducing grouping errors and improving prediction accuracy.

3) Strong adaptability. Fuzzy entropy works well with data series of different lengths and characteristics. It can adapt to the varying fluctuation patterns and specificities of pork, beef, and mutton price series, aiding in the construction and optimization of AM-LSTM prediction models.

3.AM-LSTM is not a very new algorithm. Both AM and LSTM have been used in several different research efforts. And their integration is very common.

Response:

Thank you for your insightful comment regarding our choice of AM-LSTM for price prediction. We selected AM-LSTM for the following reasons:

1) Characteristics Match. AM-LSTM combines the attention mechanism for locating key price series information and LSTM for capturing long-term trends, making it suitable for nonlinear, non-stationary, multi-scale price data.

2) Innovative Approach. We enhanced AM-LSTM by preprocessing with CEEMD and fuzzy entropy, refining features with fuzzy information granulation, and optimizing based on price data characteristics.

3) Experimental Validation. AM-LSTM outperformed other algorithms in predicting pork, beef, and mutton prices, verifying its applicability and effectiveness.

4. Many integrated methods for RUL prediction are not discussed in the work. Such as VMD, PF and GPR are integrated to the battery RUL prediction. BLS and LSTM integrated to the battery RUL prediction. PSO, RVM and voltage for analog circuit fault prognostics, etc.

Response:

Thanks very much for the constructive advice. We agree that ensemble methods are important in RUL prediction. But our study focuses on forecasting pork prices, which is a complex problem due to its nonlinearity and non-stationarity. These algorithms may encounter different challenges when addressing such problems. So we have expanded our introduction to discuss these methods and justify our choice of CEEMD, FE, FIG, and AM-LSTM for livestock price series interval forecasting, and updated our references.

The fluctuation of livestock product prices exhibits high nonlinearity and instability, being intricately influenced by a multitude of complex factors such as market supply and demand, production costs, and policy regulations. This results in a high degree of complexity and uncertainty in the related data, posing a significant challenge for accurate prediction [5]. Statistical and artificial intelligence techniques are the two main approaches used to anticipate the products price [6-8]. auto regressive moving average (ARMA) Traditional statistical methods have difficulty with the highly nonlinear and unstable nature of animal commodity price data series. Data-driven models based on machine learning (ML) have emerged for agricultural product prices prediction thanks to their excellent mapping and self-learning capabilities. Many research has applied traditional Machine Learning (ML) models, including autoregressive models, back propagation neural network (BPNN), extreme learning machine (ELM) and random forest, in the field of product price prediction, encompassing livestock products, major crops, and other commodities [9-12]. Jin et al. [13] developed a Gaussian process regression (GPR) model to forecast wholesale prices of yellow corn. Zhou et al. [14] employed an optimized relevance vector machine (RVM) to accurately predict the prices of three precious metals: silver, palladium, and platinum. However, these models often encounter issues such as large prediction errors and insufficient adaptability when dealing with complex dynamic changes and large-scale datasets. Given that product prices evolve dynamically over time, there is a complex interplay between current prices and historical time series data, which profoundly reveals the causality and correlation between time series. To effectively capture these complex relationships, deep learning (DL) technologies have emerged, such as convolutional neural networks (CNN) and long short-term memory (LSTM) models, which can significantly improve the temporal dependence of time series data and capture long-term dependencies within it. Numerous studies have demonstrated the outstanding performance of the DL mode in livestock product price prediction. Nayak et al. [15] had shown that ML algorithms outperform conventional statistical methods in predicting the prices for essential crops like tomatoes, onions, and potatoes in major India markets. Gu et al. [16] proposed an innovative dual input attention LSTM model, which demonstrated high efficiency in predicting agricultural product prices. A Hidden Markov (HM) had been combined with six baseline DL models, Recurrent Neural Networks (RNN), CNN, LSTM, Gated Recurrent Units (GRU), Bidirectional LSTM and Bidirectional GRU to predict the nonlinear and nonstationary price data of agricultural commodities in [17]. Harshith et al. [18] demonstrated that time series analysis models such as Deep Neural Networks (DNN), RNN, LSTM, and GRU, are highly effective in forecasting the daily cumin prices of Unjha market, Gujarat, India. Furthermore, these deep learning techniques are also widely applied in the prediction of intrusion detection systems. These deep learning algorithms, especially LSTM [5], demonstrate robust capabilities in time series prediction, capable of finely interpreting temporal information and utilizing it to identify deep causal relationships hidden within data sequences. When dealing with datasets that are highly dependent on time, this not only enhances sensitivity to time series features, but also significantly improves the model's predictive power and accuracy by maintaining the temporal integrity of the data and capturing subtle temporal variations. However, LSTM also exhibits strong dependencies on feature selection and the configuration of model hyperparameters. Many meta-heuristic algorithms, such as particle swarm optimization (PSO) [19], crow search algorithm (COA) [5], and differential evolution (DE) [20], are often used to optimize LSTM for hyperparameter tuning, weight initialization, and other tasks. Yet, when processing large-scale datasets, these algorithms face challenges such as high computational resource consumption, complex parameter adjustment, and susceptibility to local optimization. The introduction of attention mechanisms (AM) in LSTM models [21], however, can dynamically focus on key information within the sequence, enhancing the model's attention to important features and improving the accuracy and flexibility of sequence modeling and prediction. This innovation has garnered widespread attention in the field of optimization. In order to increase the flexibility and precision of time series prediction models in a diverse data source environment, data decomposition technology has emerged as a crucial tactic. This approach reduces the difficulty of prediction tasks by using preprocessing of complicated and fluctuating time series to uncover simpler and more readable signal patterns within the data [22, 23]. Data decomposition techniques mainly include wavelet packet transform [24], empirical mode decomposition (EMD) [25-27], variational mode decomposition (VMD) [28], and other variants. Fu et al. [29] provided a robust prediction approach for pig prices based on ensemble empirical mode decomposition (EEMD) and LSTM. Li et al. [30] adopted a combination of the VMD and GRU neural network to predict the prices of beef, mutton, and pork while taking into account the effects of heterogeneous non-time series on livestock products price, such as the variety, growth cycle, longitude and latitude. Sun et al. [31] introduced a comprehensive prediction model that combined VMD, EEMD, and LSTM to forecast the average wholesale weekly price for pork, Chinese chives, shiitake mushrooms, and caulif

---

## [Editor Report · Decision Letter 2]

22 Jan 2025

Interval price prediction of livestock product based on fuzzy mathematics and improved LSTM

PONE-D-24-41802R2

Dear Dr. Peng,

We’re pleased to inform you that your manuscript has been judged scientifically suitable for publication and will be formally accepted for publication once it meets all outstanding technical requirements.

Kind regards,

Jinran Wu, PhD

Academic Editor

PLOS ONE

Additional Editor Comments:

Congratulations!

---

## [Editor Report · Acceptance letter]

PONE-D-24-41802R2

PLOS ONE

Dear Dr. peng,

I'm pleased to inform you that your manuscript has been deemed suitable for publication in PLOS ONE. Congratulations! Your manuscript is now being handed over to our production team.

Kind regards,

on behalf of

Dr. Jinran Wu

Academic Editor

PLOS ONE